# Microscopic and Transcriptomic Analysis of Pollination Processes in Self-Incompatible *Taraxacum koksaghyz*

**DOI:** 10.3390/plants10030555

**Published:** 2021-03-16

**Authors:** Tassilo Erik Wollenweber, Nicole van Deenen, Kai-Uwe Roelfs, Dirk Prüfer, Christian Schulze Gronover

**Affiliations:** 1Institute of Plant Biology and Biotechnology, University of Muenster, Schlossplatz 8, 48143 Muenster, Germany; wollenweber@uni-muenster.de (T.E.W.); nicole.vandeenen@uni-muenster.de (N.v.D.); dpruefer@uni-muenster.de (D.P.); 2Fraunhofer Institute for Molecular Biology and Applied Ecology IME, Schlossplatz 8, 48143 Muenster, Germany; kai.uwe.roelfs@ime.fraunhofer.de

**Keywords:** self-incompatibility, Asteraceae, *Taraxacum koksaghyz*, microscopy, RNA-Seq, digital gene expression

## Abstract

The transition of the Russian dandelion *Taraxacum koksaghyz* (Asteraceae) to a profitable, alternative crop producing natural rubber and inulin requires the optimization of several agronomic traits, cultivation conditions and harvesting procedures to improve the yield. However, efficient breeding is hindered by the obligatory sexual outcrossing of this species. Several other asters have been investigated to determine the mechanism of self-incompatibility, but the underlying molecular basis remains unclear. We therefore investigated the self-pollination and cross-pollination of two compatible *T. koksaghyz* varieties (TkMS2 and TkMS3) by microscopy and transcriptomic analysis to shed light on the pollination process. Self-pollination showed typical sporophytic self-incompatibility characteristics, with the rare pollen swelling at the pollen tube apex. In contrast, cross-pollination was characterized by pollen germination and penetration of the stigma by the growing pollen tubes. RNA-Seq was used to profile gene expression in the floret tissue during self-pollination and cross-pollination, and the differentially expressed genes were identified. This revealed three candidates for the early regulation of pollination in *T. koksaghyz*, which can be used to examine self-incompatibility mechanisms in more detail and to facilitate breeding programs.

## 1. Introduction

The Asteraceae is one of the largest families of angiosperms [1] and ~63% of aster species are incapable of self-fertilization due to self-incompatibility (SI) [2]. Among the remaining families, ~10% are considered partial (or pseudo) self-incompatible (PSI) and ~27% are self-compatible (SC). The Russian dandelion *Taraxacum koksaghyz* is a diploid (2n = 16) member of the Asteraceae that has been investigated as an alternative source of natural rubber and inulin [3,4,5,6]. However, undomesticated plants from wild collections are extremely heterogeneous in terms of agronomic performance. Various traits (such as root morphology) and practices (such as cultivation time and conditions, sowing time and harvesting processes) have been exploited to increase the yield of natural rubber and inulin [7,8,9]. Furthermore, genomic simple sequence repeat markers are available for use in future genomic breeding programs [10].

Although there are many apomictic species in the genus *Taraxacum* [11], *T. koksaghyz* is a SI outcrossing species [12], making it difficult to obtain homozygous lines for hybridization [3]. Intercrossing such hybrids could enhance natural rubber and inulin production via hybrid vigor, which is a widespread crop breeding strategy [13]. SC plants enable the crossing of otherwise non-crossable cultivars (expressing the same *S*-haplotype) to generate progeny from selfing [14]. Furthermore, they are easier to maintain and evaluate, and are less dependent on insect pollination, as shown by comparing SC domestic and SI wild-type cultivars of sunflower (*Helianthus annuus*) [15]. Understanding the reproductive strategy of *T. koksaghyz* could be therefore beneficial to transform this wild species into a useful crop as well as to shed light on the SI mechanism in Asteraceae.

The SI mechanism that promotes outbreeding in angiosperms is similar to pathogen defense (especially against fungi) and several defense components may have been recruited to the SI mechanism [16,17]. There are two major SI mechanisms in angiosperms: gametophytic SI (GSI), found for example among the Solanaceae and Rosaceae [18] and regarded as the more ancient SI mechanism [19], and sporophytic SI (SSI), found for example among the Brassicaceae and also considered as the mechanism deployed by the Asteraceae [20].

The first interaction between pollen and pistil occurs rapidly during the adhesion of pollen to papilla cells, and the underlying mechanism during compatible pollination in Arabidopsis has been recently reviewed [21]. The exine-embedded pollen coat forms an attachment foot with the proteinaceous pellicle, which is needed to accept compatible pollen [22]. The reorganization of the pollen coat (coat conversion) allows water to enter from the stigma and the influx of extracellular Ca^2+^, leading to pollen rehydration and metabolic activation [22,23]. The nonpolar state of the pollen grain changes shortly after hydration to a highly polarized state, in which the cytoplasm and cytoskeleton are reorganized to allow the deposition of targeted secretory vesicles and callose [24,25]. The pollen tube emerges through one of the apertures, grows through the attachment foot and arrives at the papilla cells [26]. This polar growth requires crosstalk between the pollen and stigma as well as fine-tuned cell wall modification. The pollen tube cell wall can be divided in two parts: the shank and tip. The shank is more rigid, with a primary and secondary cell wall. The primary cell wall incorporates polysaccharides, such as cellulose, pectins and xyloglucan, whereas the secondary cell wall is almost exclusively composed of callose. The softer tip lacks a secondary cell wall and the primary cell wall is mostly composed of pectins, allowing directional changes during pollen tube expansion and growth [27,28]. The cell wall polymers are mostly synthesized by the Golgi apparatus and secreted via vesicles to the plasma membrane at the tip. Golgi-derived polymers include pectins, hemicelluloses, and hydroxyproline-rich glycoproteins such as arabinogalactan proteins (AGPs) and extensins (EXTs) [29]. Other cell wall polymers such as cellulose and callose are synthesized directly at the plasma membrane [29]. 

When the pollen tube arrives at the surface of the stigma, it enters the papilla and grows beneath the papilla cell wall into deeper transmitting tissue and finally through the style to the ovary [30]. The pollen is guided by numerous attractants mainly originating from the ovary, and these are sensed by multiple receptor-like kinases [31,32]. A gradient of the phytohormone *γ*-aminobutyric acid (GABA) leading up to the micropyle is also involved in pollen tube guidance [33]. At the ovule, the pollen tube releases its sperm cells, leading to double fertilization in angiosperms [34]. Following fertilization, ethylene signaling is induced to abolish further pollen tube attraction to the ovary [35].

Several early pollination events have been confirmed in asters such as *H. annuus* and the ragwort *Senecio squalidus*, including: (1) the adhesion of the pollen to the papilla (probably dependent on pollen exine [36]); (2) the fusion of aliphatic stigma surface material, combined with stigmatic secretion and pollenkitt to an attachment foot as well as pollen hydration; (3) the germination of the pollen tube from the apertures; (4) pollen tube growth through the attachment foot and the papillae and, subsequently, penetration of the stigma at the base of papilla cells; and (5) intracellular growth of the pollen tube through the stigma toward the style and ovary [37,38]. When pollen is applied to the papillae of *Zinnia elegans* and *H. annuus*, microscopy revealed rapid pollen tube growth and penetration of the stigmatic surface within 10–30 min after compatible pollination [38,39].

To our knowledge, detailed microscopic studies of pollen–stigma interactions and pollination in *T. koksaghyz* have not been reported thus far. Previous studies have shown morphological differences between *T. officinale* and *T. koksaghyz* in terms of flower and seed structure [40], and have cataloged the morphological appearance of the *T. koksaghyz* capitulum and floral organogenesis in more detail [41]. Accordingly, we studied the pollen–stigma interaction mechanism in *T. koksaghyz* after incompatible and compatible pollination and investigated the gene expression profiles during pollination and pollen tube growth.

## 2. Results

### 2.1. Seed Development after Self-Pollination and Cross-Pollination of T. koksaghyz

To evaluate the differing morphological and molecular processes that accompany self-pollination (SP) and cross-pollination (CP) in *T. koksaghyz*, we used two compatible varieties named TkMS2 and TkMS3 for crossing experiments. To document overall flower and seed morphology after SP and CP, flowers from each variety were pollinated manually and the tissues were examined for the development of infertile and fertile seeds after 12–14 days. Neither variety showed evidence of fertile seed development following manual SP after the onset of flowering (Figure 1a, TkMS2 as a representative example) and the infertile seeds are shown in Figure 1b (TkMS2 as a representative example). In contrast, the CP of TkMS2 and TkMS3 in both directions led to the development of a typical, fanned-out pappus (Figure 1c) and a mixture of fertile and infertile seeds (Figure 1d). To determine the number of fertile seeds after manual CP, nine independent pollinations were carried out with TkMS3 as the pollen donor and seven with TkMS2 as the pollen donor. We then counted the total number of florets within each capitulum (Figure 1e) and the relative number of fertile and infertile seeds harvested from the complete capitulum after 12–14 days (Figure 1f).

The infertile seeds produced by SP and CP were flat and whitish in color (Figure 1b,d) but the SP seeds were shorter and irregular in shape (Figure 1b). In contrast, the fertile seeds were gray–brown in color, solid in structure, and the shape was conical with upward-bending spinules (Figure 1d). 

The number of florets within the capitulum of *T. koksaghyz* varieties TkMS2 and TkMS3 varied considerably, ranging from 48 to 96 TkMS2 florets and from 68 to 124 TkMS3 florets (Figure 1e). However, the mean relative proportion of harvested fertile seeds (representing successful pollination) was similar in both CPs, with 13.0% for TkMS2xTkMS3 and 12.2% for TkMS3xTkMS2 (Figure 1f). The low proportion of fertile seeds reflects the incomplete manual pollination of the large number of florets.

### 2.2. Evaluation of Pollen Tube Growth by Microscopy

As an outcrossing member of the Asteraceae, *T. koksaghyz* presumably features an SSI system based on the recognition and inhibition of SI pollen at the stigma. In contrast, pollen germination and pollen tube growth should be supported after compatible CP. To evaluate pollen recognition and pollen tube growth in more detail, the stigma was analyzed by microscopy. Individual *T. koksaghyz* florets from the CP TkMS2xTkMS3 (fertile seed production) and SP TkMS2 (infertile seed production) were harvested at different time points after pollination, and the pollen on the stigma was visualized with aniline blue staining (Figure 2).

Each floret was composed of five united petals represented by the corolla, as well as five fused stamens building the anther tube [41]. Following the onset of flowering, the stigma grows through the anther tube to expose the receptive site of the forked stigma for potential pollinators, carrying the pollen at the outer cuticula. Analysis of the receptive site by microscopy revealed fewer adhered pollen on the papilla cells after SP compared to CP at different time points, indicating the detachment of pollen due to insufficient adhesion. Although pollen germination occurred after SP at the papilla cells, only limited pollen tube growth was observed, with no penetration of the stigma (Figure 2b,c). The few germinating pollen tubes showed either strong aniline blue staining on the swollen pollen tube wall (Figure 2b, white arrows) or swelling of the pollen tube apex (Figure 2c) with the width increasing exemplarily from 5.2 µm to 9.4 µm at the papilla cells (double-headed arrows). 

In contrast, pollen tubes generated following CP were detected after penetrating the stigma (Figure 2d,f). These pollen tubes grew in the direction of the style (white arrows) but no pollen tubes were observed with or without prior lactic acid fixation in the inferior part of the stigma or at the ovary. Several pollen tubes growing simultaneously between papilla cells were observed within 40 min after pollination (Figure 2e). However, not all of the adherent pollen after CP resulted in the successful germination of a pollen tube: the two stigma-penetrating pollen tubes in Figure 2d are accompanied by three pollen grains with no germinating pollen tubes. The pollen from TkMS2 or TkMS3 was indistinguishable, suggesting that self-pollen might have attached to the stigma during manual pollination and was rejected. Signals of varying intensity representing different warty parts of the exine were observed in different layers of the pollen (Figure 2b–d).

### 2.3. Transcriptomic Analysis of CP and SP

Various processes ensure the recognition of compatible pollen and successful double fertilization in angiosperms. Several genes that participate in the SSI recognition process in brassicas have been identified, but aster genes associated with pollen recognition and inhibition are not well characterized. To identify differentially expressed genes (DEGs) in *T. koksaghyz* during CP and SP, TkMS2 and TKMS3 plants were crossed with each other or manually self-pollinated, and the upper part of the style, the stigma, and the upper part of the anther tube were harvested for RNA-Seq analysis. To identify DEGs at different time points during CP and SP, tissues from five florets harvested 10, 30 and 60 min after pollination were pooled as T1, and tissues harvested 120 and 240 min after pollination were pooled as T2. We also harvested tissue prior to pollination (pooled as T0). The florets from eight independent pollinations were used for each CP (TkMS2xTkMS3 and TkMS3xTkMS2) as well as the florets of 2–6 capitula for each SP (TkMS2 and TkMS3). 

#### 2.3.1. Filtering and Venn Diagrams 

The reads were mapped against the *T. koksaghyz* reference genome [5] with 80–83% efficiency. Among the 47,230 detected transcripts, 29,699 (62.9%) could be annotated (Figure 3a). Transcripts were considered to show significant differential expression between the T0, T1 and T2 pools based on a log_2_ fold-change of ≥1 or ≤–1 in normalized expression and a Benjamini-Hochberg false discovery rate (FDR)-corrected *p*-value ≤ 0.05. Subsequent filtering resulted in 1040 significant DEGs with 849 (81.6%) annotated. 

To narrow down candidate genes specifically participating in pollination, we applied two different filtering strategies. First, we filtered transcripts with significant differential expression within the CP and SP samples of TkMS2 or TkMS3 at different time points. This identified 158 transcripts including 147 annotated genes that were visually analyzed by comparing the expression profiles within each variety. The search for candidates using this approach focused more on the individual plants than the comparison of varieties, and yielded three annotated genes showing a promising expression profile (Figure 3a, bottom left). The candidates were selected based on the observation of differing expression profiles between CP and SP in the T0 vs. T1 and T1 vs. T2 comparisons, coupled with significant differences in log_2_ fold-change values within the CP and SP samples.

The second filtering approach involved the comparison of expression profiles within the two plant varieties and among the pooled time points (T1 vs. T0, T2 vs. T1, or T2 vs. T0). This identified 543 transcripts including 515 annotated genes with the same expression profile (upregulation or downregulation) in TkMS2 and TkMS3 during CP or SP at all time points. Venn diagram analysis allowed the selection of DEGs with opposing expression profiles in CP and SP (Figure 3b,c). This yielded two annotated genes (utg21288.4 and utg8190.13) showing opposing expression patterns (upregulated during CP, downregulated during SP) in the T1 vs. T0 comparison (Figure 3b, bold underlined). 

Subsequent BLASTX searches against the NCBI non-redundant (nr) database revealed that utg21288.4 showed 82.67% identity to leucine-rich repeat extension-like protein 4 (LRX4) and utg8190.13 showed 99.55% identity to the tubulin β chain-like protein (TUBB) both from lettuce, *Lactuca sativa* (Table 1). We also found 73 DEGs sharing the same upregulated expression during CP and SP shortly after the pollen-pistil interaction (Figure 3b) but only one DEG (utg8071.58) sharing the same downregulated expression profile in CP and SP when comparing T1 vs. T0. This was annotated as a protein of unknown function (DUF4228) from the globe artichoke, *Cynara cardunculus* var. *scolymus* (Figure 3b). The number of downregulated DEGs was small (only 20 in CP and 45 in SP) compared to the upregulated genes (105 in CP and 132 in SP). Therefore, 21.5% of DEGs in the T1 vs. T0 comparison were downregulated whereas 78.5% were upregulated.

Although the T1 vs. T0 comparison revealed two DEGs (*LXR4* and *TUBB*) with opposing expression profiles, no such DEGs were detected in the T2 vs. T1 comparison (Figure 3c). Most of the DEGs were exclusively present within one group (CP or SP, upregulation or downregulation), with only 24 DEGs common to the upregulated CP and SP genes and 16 DEGs common to the downregulated CP and SP genes. 

Visual analysis of the DEGs identified during CP and SP in either plant variety revealed an additional gene (utg9900.15) with opposing expression profiles, showing significant log_2_ fold-changes during the CP and SP of TkMS3 and overall downregulation from T0 to T2 during the SP of TkMS2 (Figure 3a, bottom left panel). This DEG showed 91.77% identity with a candidate xyloglucan endotransglucosylase/hydrolase 33 (*XTH33*) from lettuce (Table 1).

The high proportion of SP reactions in both SP and CP tissues, reflecting the overall low proportion of successful pollination events (Figure 1f), combined with the stringent filtering approach, resulted in the detection of upregulated DEGs solely during CP.

#### 2.3.2. Digital Gene Expression Profile of Selected DEGs

The filtered and selected DEGs resulting from the transcriptomic analyses (Figure 3) were visualized based on the detected transcript levels relative to T0 (Figure 4).

Although the log_2_ fold-changes were not in all cases significant in TkMS2 and TkMS3, the expression profiles and transcript levels were comparable. CP triggered the upregulation of all three transcripts from T0 to T1 and subsequent downregulation from T1 to T2, whereas SP resulted in the downregulation of all three transcripts from T0 to T1 and subsequent downregulation (or unchanged expression) from T1 to T2 (Figure 4). 

#### 2.3.3. Gene Ontology Analysis of DEGs in the T1 vs. T0 and T2 vs. T1 Comparisons

To investigate the direct effect of pollination and categorize the DEGs by function, we identified and compared the overrepresented Gene Ontology (GO) terms. We focused on the comparison between DEGs present during CP and SP in either T1 vs. T0 or T2 vs. T1 (summed for TkMS2 and TkMS3) against the GO background with 29,699 identified genes (independent of specific time points) (Figure 3a, top panel). Enriched GO terms were selected with a *p*-value ≤ 0.001 and were depicted for each subset of the DEGs included in the Venn diagrams (Figure 3b,c) against the GO background (Figure 5).

The largest number of enriched GO terms was found for the biological process domain, with 11 GO terms in the T1 vs. T0 comparison, representing 20 CP downregulated, 45 SP downregulated, 105 CP upregulated and 132 SP upregulated genes (Figure 5a). An additional eight enriched GO biological processes were observed in the T2 vs. T1 comparison, representing 57 CP downregulated, 84 SP downregulated, 44 CP upregulated and 92 SP upregulated genes (Figure 5b). Most of these GO terms were connected to response processes. Additional GO terms associated with signaling (GO:0023052 and GO:0007165), multi-organism processes (GO:0051704) and localization (GO:0051179) were also present in the T1 vs. T0 comparison (Figure 5a), whereas regulation of biological quality (GO:0065008) and nitrogen compound metabolic process (GO:0006807) were enriched in the T2 vs. T1 comparison (Figure 5b). Although the T2 vs. T1 comparison also revealed several enriched GO terms in the cellular component domain, specifically intracellular organelle part (GO:0044446), organelle part (GO:0044422), intracellular (GO:0005622) and intracellular part (GO:0044424), none of these categories was enriched in the T1 vs. T0 comparison. Among molecular functions, enriched categories included transcription activity-related GO terms (GO:0140110 and GO:0003700) in the T1 vs. T0 comparison (Figure 5a) as well as transmembrane transporter activity (GO:0022857) and binding (GO:0005488) in the T2 vs. T1 comparison (Figure 5b).

The relative abundance of genes in the T1 vs. T0 comparison showed a strong enrichment for upregulated DEGs during CP and SP but the depletion of downregulated DEGs during SP regarding the GO terms response to stimulus (GO:0050896; 44%, 42% and 18%), response to stress (GO:0006950; 32%, 29% and 9%) and response to chemical (GO:0042221; 34%, 31% and 4%) relative to the GO background (28%, 17% and 13%, respectively) (Figure 5a, Appendix A). The subset of downregulated DEGs during CP showed no strong differences to the GO background (30%, 15% and 15%, respectively). However, the GO terms response to stimulus and response to chemical in the T2 vs. T1 comparison were 10–15% more abundant compared to the GO background and were associated with DEGs in all subsets (Figure 5b, Appendix A). The subsets of GO terms in the T1 vs. T0 comparison associated with upregulated DEGs during CP and SP were more abundant compared with the GO background, whereas GO terms associated with downregulated DEGs in CP and SP were either missing completely or missing from among the transcription activity-related GO terms. The abundance of DEG subsets within the cellular component domain in the T2 vs. T1 comparison either decreased or remained similar to the GO background (Figure 5b). 

## 3. Discussion

The development of fertile or infertile seeds in SI angiosperms is the final checkpoint to determine whether CP has succeeded or failed. However, fertile seed development is only possible if SSI or GSI is overcome, allowing compatible pollen or pollen tubes to be accepted, facilitating the pollination process [18]. In our heterogeneous *T. koksaghyz* varieties, we found that TkMS2 and TkMS3 achieved successful manual CP (Figure 1b,d) but SP did not result in fruit set (Figure 1a,c). The mean proportion of fertile seeds was 13.0% in TkMS2 (n = 9 independent crossings) and 12.2% in TkMS3 (n = 7 independent crossings), indicating manual pollination of both varieties was comparable in efficiency (Figure 1e,f). Asteraceae are regarded as SI when manual SP achieves < 0.05% fruit set [2]. Approximately 10% of Asteraceae species show partial SI [2], and the absence of fertile seed development in SP TkMS2 and TkMS3 suggests that *T. koksaghyz* is also a SI species. 

We carried out the first microscopic analysis of SP and CP in *T. koksaghyz*. Many pollen grains from the donor TkMS3 were present on the stigmatic papillae of TkMS2, but few self-pollen were observed on the stigma. The few pollen grains we detected were able to germinate but not penetrate the stigma after 1 h or 24 h (Figure 2b,c). The stigma does not support incompatible pollen leading to the suspected failure of pollen grain adhesion, so most self-pollen grains were washed away during aniline blue staining. Similar results were reported for the SP of SI *Brassica napus* line W1 [42]. The infrequent germination of self-pollen was also observed in *H. annuus*. However, the germinated pollen tubes grew parallel to the papillae and failed to penetrate the stigma [38]. The swelling of the pollen tube tip and accumulation of aniline blue in the apex of germinated self-pollen supports observations in *S. squalidus* [37,43]. However, callose plug formation in the papilla cells near self-pollen was not as prominent as was previously shown in *S. squalidus* [37,43,44].

The compatible pollination with TkMS3 as the donor resulted in several pollen grains adhering to the stigma of TkMS2, with traceable pollen germination and pollen tube growth within 30 min (Figure 2d–f). The pollen tubes could be detected for a short distance in stigmatic tissue beneath the papilla cells and growing along the stigma to the style, probably reaching the inferior ovary. However, pollen tubes were not traceable within the style or ovary using aniline blue, as reported for other *Taraxacum* species [45]. Aniline blue staining of *T. japonicum* and *T. longeappendiculatum* resulted in limited traceability within regions between the stigma and style, style and ovary, and around the ovules. An additional clearing step with lactic acid fixation did not achieve the visualization of pollen tubes within the style (Figure 2d,e), as previously shown for *Syzygium* sp. [46]. Although the absence of pollen tubes after SP and the presence of pollen tubes after compatible CP within the style indicates SSI [44,47], the strongest evidence for an SSI mechanism is the absence of pollen on the papilla cells, presumably due to reduced pollen germination rate, and the tip swelling of pollen tube apex after SP before reaching the style (Figure 2b,c) as it is also described in *S. squalidus* [37,43]. In contrast, in plants deploying the GSI mechanism, such as Solanaceae and Rosaceae, pollen tubes are inhibited within the stylar transmitting tract [48]. Further data on pollen tube growth within the style of *T. koksaghyz* could be gained from transgenic plants expressing a fluorophore gene driven by a pollen-specific promoter [49].

Comparative transcriptomics was used to analyze the pollination process at the molecular level. Given the potential for additional SP during manual CP (Figure 1f), our method focused on DEGs upregulated during CP. Although mechanical emasculation of flowers before the pistillate stage, in which the stigma reveals the receptive papillae, could avoid additional SP during CP, the emasculation of *T. koksaghyz* florets is not feasible. DEGs modulated by CP regardless of potential interactions with the regulation of SP on the same stigma could reveal molecular processes involved in *T. koksaghyz* pollination. Among the DEGs involved in the immediate response to CP, two showed a promising expression pattern (utg21288.4 and utg8190.13) and were narrowed down by constructing a Venn diagram (Figure 3b, bold and underlined). BLASTX searches against the NCBI nr database indicated similarities to lettuce *LRX4* and *TUBB*, respectively. 

The LRX family is involved in the maintenance of pollen tube cell wall integrity and assembly [50,51]. LRX proteins are EXTs containing two or more EXT motifs (Ser-Pro_(3-5)_). Other proteins with at least two EXT motifs include proline-rich EXT-like receptor kinases (PERKs), most group I formins (with additional actin-microtubule binding domains), extracellular hybrid EXTs including the hybrid AGP-EXT and LRX, and conventional EXTs, the most abundant proteins with EXT motifs encoded in the Arabidopsis genome. All but the first group contain a signal peptide and several are known to be involved in pollen tube and root hair tip growth [52]. LRX proteins also play a significant role in plant defense and specific protein–protein interactions [53]. Host–pathogen and pollen–stigma interactions share similar recognition processes [54,55]. Accordingly, the involvement of LRX4 in pollination is not surprising.

The DEG annotated as *TUBB* (β-tubulin) is likely to be a fundamental driver of morphogenesis [56]. Tubulins belong to an ancient protein family that forms a structural constituent of the cytoskeleton as well as facilitating microtubule polymerization during processes such as organelle transport [57] and cell expansion [58]. Several β-tubulin genes in flax (*Linum usitatissimum*) represent different tubulin subpopulations involved in diverse microtubule functions, such as cell elongation, cell wall thickening and pollen tube growth [56]. The *T. koksaghyz TUBB* gene may therefore participate in microtubule polymerization within pollen tubes. Specific roles in organelle transport or cell expansion would require further investigation.

We also identified the DEG utg9900.15, with high sequence identity to *XTH33*, a lettuce gene encoding xyloglucan endotransglucosylase/hydrolase involved in cell wall biogenesis. Plant cell expansion is driven by the incorporation of new cell wall material, not by thinning of the existing cell wall [59]. Xyloglucan is the major hemicellulosic polysaccharide within the primary cell wall, connecting adjacent cellulose microfibrils, and was also shown to control cell expansion in the pollen tube cell wall [60]. Xyloglucan possesses a 1,4-β-glucan backbone with 1,6-α-xylosyl residues. Additional side chains can contain fucose, which is typical for primary cell wall xyloglucans, and galactose [61]. XTHs cleave xyloglucan and re-join the ends to different substrates, such as water or the non-reducing terminus of available xyloglucan chains or oligosaccharides, changing the properties of the cell wall [62]. Furthermore, the fucosylated xyloglucan content of the inner cell walls of pollen tubes was depleted in the Arabidopsis mutant *lrx8 lrx9 lrx11* [50]. Along with *LRX4* and *TUBB*, *XTH33* may therefore be involved in the *T. koksaghyz* pollination process, more specifically in pollen tube development by maintaining the integrity of the cell wall.

Among the DEGs with the same expression profile in TkMS2 and TkMS3, most were upregulated between T0 and T1 during both CP and SP. The abundance of upregulated DEGs in the early phase (20–30 min) after both types of pollination was also shown by time-course transcriptome analysis in the self-compatible *B. napus* line Westar and the self-incompatible line W-3, corresponding to the completion of pollen–stigma interactions within this time [63]. In *T. koksaghyz*, we identified 327 upregulated and 65 downregulated DEGs in the T1 vs. T0 comparison, and this may similarly indicate the early activation of biological processes by pollination and finally contribute to SI.

For example, the downregulated DEG utg8071.58 within SP and CP of T1 vs. T0 which was annotated as a protein of unknown function (DUF4228). Other members of the domain of unknown function (DUF) family have already been implicated in the process of SI mechanism. DUF247 was shown to co-segregate with the multiallelic *S*-locus and potentially plays a major role as the male determinant within GSI of perennial ryegrass [64]. Sequence similarities between the DUF247 of perennial ryegrass and the DEG utg8071.58 within this study were only marginal present (14.7% amino acid sequence similarity), indicating a different role of the DEG within *T. koksaghyz*. However, since utg8071.58 was found to be exclusively downregulated, its role during the pollination process will be integrated in future studies.

Pollen-specific upregulated genes may contribute to the upregulated DEGs, supported by the analysis of GO terms in the T1 vs. T0 comparison. Most of the enriched GO terms were connected to biological processes such as response to stimulus (GO:0050896), response to stress (GO:0006950) and response to chemical (GO:0042221) (Figure 5a). Furthermore, GO terms related to other response processes, such as response to endogenous stimulus (GO:0009719) or biotic stimulus (GO:0009607), were enriched among the upregulated DEGs linked to CP and SP. We also found that transcription-related GO terms were enriched among the upregulated DEG linked to CP and SP, paired with the decrease or absence of downregulated DEGs within these subsets. These results indicate an overall upregulation-dependent complex response after compatible or incompatible pollination, including profound changes in DEGs related to transcriptional activity. Although we observed differences in the relative abundance of upregulated and downregulated DEGs associated with the GO terms response to stimulus and response to chemical in the T1 vs. T0 comparison, all subsets showed an equal increase in the number of DEGs related to these GO terms in the T2 vs. T1 comparison. This suggests that the upregulated and downregulated DEGs play an important role during pollination regardless of compatibility. Furthermore, a significant decrease in the number of DEGs associated with the cellular component domain was observed, which implies these genes play a more important role during the earlier stages of pollination.

Our results help to define the SI mechanism and identify molecular components of the pollination process in *T. koksaghyz*, an industrially relevant plant for the production of natural rubber and inulin. Microscopy provided strong evidence that SI pollination reflects an SSI mechanism, and confirmed the penetration of the stigma within the first 30 mins after compatible pollination. Furthermore, the analysis of DEGs revealed three candidate genes (*LRX4, TUBB* and *XTH33*) that may drive pollen tube growth in *T. koksaghyz*, and these will be characterized in more detail in future studies. GO term analysis of DEGs in the T1 vs. T0 comparison indicated an overall enrichment for biological processes such as signaling and various response mechanisms, as well as an increase in transcriptional regulation after compatible and incompatible pollination. These processes will also be investigated in more detail in the future. Our transcriptomic data provide a foundation for the further analysis of pollination and the SSI mechanism in *T. koksaghyz* to promote its transition from a wild plant to a domesticated crop.

## 4. Materials and Methods

### 4.1. Plant Material, Cultivation and Pollination

Clonally propagated *T. koksaghyz* wild-type plant varieties TkMS2 and TkMS3 were cultivated at 18 °C and 20 klux (600 W high-pressure sodium lamps, enhanced yellow and red spectrum) in a greenhouse with a 16-h photoperiod, as previously described [65]. Inflorescences with fully-exposed receptive sites at the forked stigmas were used for manual pollination by brushing individual flower heads of the same variety (SP) or the compatible varieties (CP) together. For transcriptomic analysis, tissue was harvested from at least five florets (anther tube and stigma, as well as the upper part of the style) before pollination and after 10, 30, 60, 120 and 240 min. We carried out eight independent CPs (TkMS2xTkMS3 and TkMS3xTkMS2), six SPs (TkMS2) and two SPs (TkMS3). 

### 4.2. Confocal Laser Scanning Microscopy (CLSM)

The upper part of the style and stigma tissue was harvested at different time points (30 min, 40 min, 60 min and 24 h after pollination) and either used directly for squash preparations or first fixed in 3:1 lactic acid/70% ethanol for 24 h then cleared with 4 M NaOH. The samples were simultaneously mounted and stained with 1:1 decolorized aniline blue and 10% glycerol.

### 4.3. RNA Extraction, RNA-Seq and Transcriptome Analysis

Total RNA was extracted using the innuPREP RNA Mini Kit (Analytik Jena, Jena, Germany) according to the manufacturer’s instructions. The absence of DNA was confirmed by PCR using the intron-spanning primers GAPDH-fwd (5’-CTTCAGAGAGATGATGTT) and GAPDH-rev (5’-CTTCCACCTCTCCAGTCCTT) followed by agarose gel electrophoreses. RNA from floret tissue was pooled as follows: RNA extracted before pollination = T0, RNA extracted after 10, 30 and 60 min = T1, and RNA extracted after 120 and 240 min = T2. A minimum of 800 ng of pooled RNA for each sample was shipped on dry ice to GATC Biotech (Constanz, Germany) where all quality control, library preparation and sequencing steps were completed prior to sequencing on the Illumina HiSeq 2500 platform (2x150 bp). Sequenced reads were trimmed to remove adapters, quality filtered and mapped against the reference genome deposited in the Genome Warehouse (http://bigd.big.ac.cn/gwh/) (20 August 2018) [5], accession number PRJCA000437. Raw read statistics can be found in Appendix A. The genome was annotated using a BLASTX search against the NCBI nr and UniProt databases, with a minimum e-value of 1.0 × 10^−5^ and a max hits value of one sequence. DEGs are recorded along with their log_2_ fold-changes and annotations (nr and UniProt) in the Appendix A.

### 4.4. Differential Expression Analysis 

The Tuxedo suite was used with default settings and transcript abundance was normalized as fragments per kilobase of transcript per million mapped reads (FPKM). Transcripts with a log_2_ fold-change of ≥1 or ≤–1 difference in normalized expression and a Benjamini-Hochberg FDR-corrected *p*-value ≤ 0.05 were defined as significantly differentially expressed.

### 4.5. GO Enrichment Analysis and Venn Diagrams 

Sets of DEGs from the T1 vs. T0 and T2 vs. T1 comparisons were plotted as Venn diagrams (http://www.interactivenn.net/) (20 November 2020) [66]. Subsets of DEG-loci used for Venn analysis can be found in Appendix A. Functional classification of the subsets was carried out using WEGO (https://wego.genomics.cn/) (27 November 2020) [67]. GO-enriched genes were identified based on the complete CP and SP subsets of all DEGs with the same expression pattern in both plant varieties compared to all GO terms related to all identified genes (time point independent) (Appendix A). Only significant differences in the enrichment of GO terms with a *p*-value ≤ 0.001 were used and subsets of upregulated and downregulated DEGs were compared (Appendix A).

## Figures and Tables

**Figure 1 plants-10-00555-f001:**
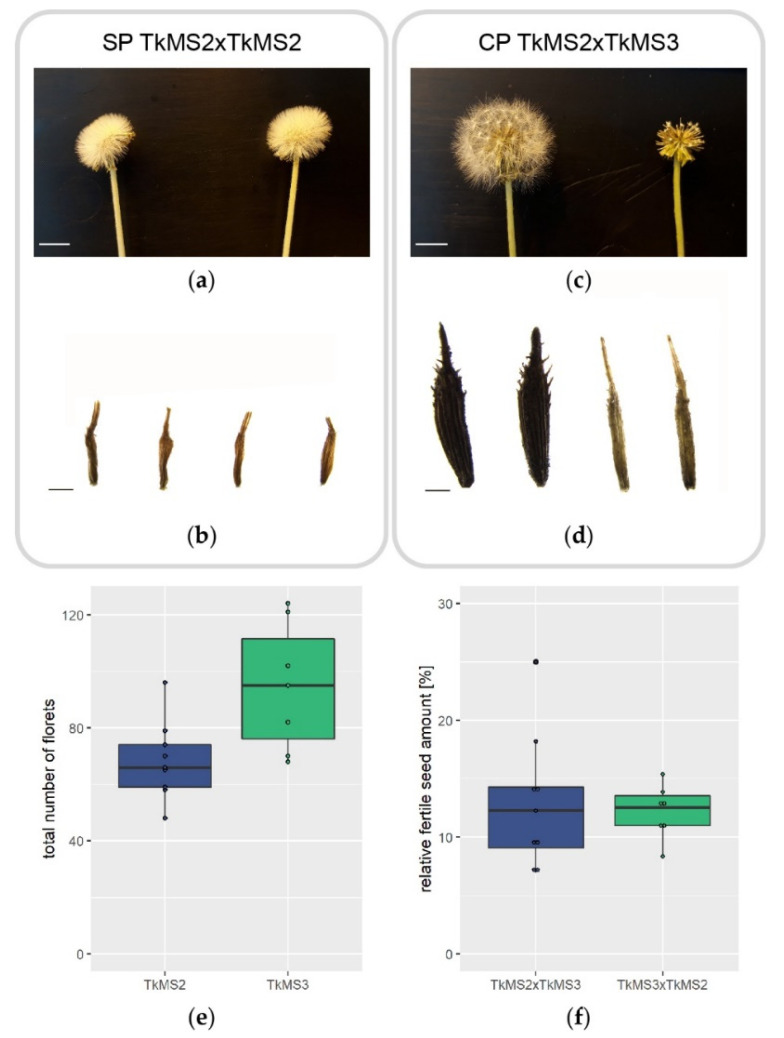
Overview of *Taraxacum koksaghyz* flower and seed morphology 12–14 days after incompatible and compatible pollination and the analysis of florets and seeds. (**a**) Following self-pollination (SP), variety TkMS2 did not produce spherical seed heads, indicating (**b**) the production infertile seeds. TkMS2 is shown as a representative example but similar results were obtained for TkMS3. The irregular shape and small size of infertile seeds was consistent with incompatible pollination. (**c**) A dandelion-typical achene was formed following cross-pollination (CP) with TkMS3 as the pollen donor. The upper part of the stalk was removed before the fertile seeds reached maturity to visualize the seeds. (**d**) Fertile (left) and infertile (right) seeds from the CP TkMS2xTkMS3. The pappus was detached prior to the formation of the seed head (8–10 days after pollination). The infertile and fertile seeds originating from the compatible pollination were similar in length but differed in shape and thickness. (**e**) The total number of florets within the flower head of TkMS2 (n = 9) and TkMS3 (n = 7) were counted and (**f**) the relative fertile seed set of compatible pollinations were calculated. Scale bars in (**a**,**c**) = 10 mm and in (**b**,**d**) = 0.5 mm.

**Figure 2 plants-10-00555-f002:**
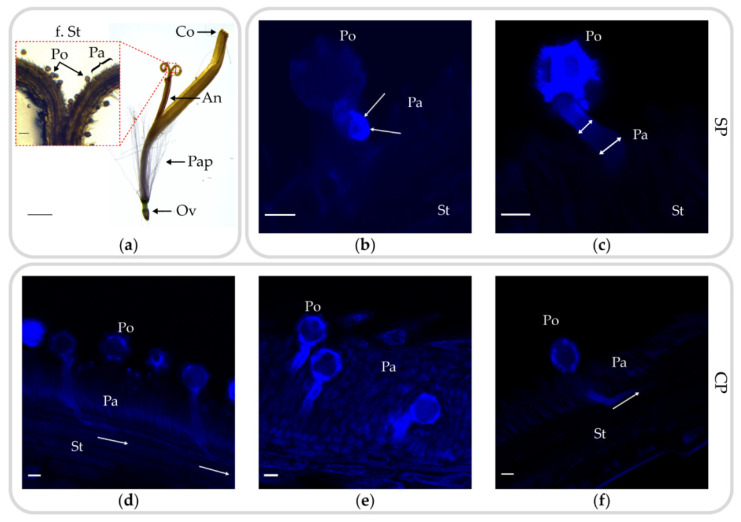
Lateral overview of a *T. koksaghyz* floret as well as incompatible (TKMS2xTkMS2) and compatible (TkMS3xTkMS2) pollen–stigma interactions revealed by confocal laser scanning microscopy. (**a**) Overview of a *T. koksaghyz* floret after the onset of flowering, showing inferior ovary (Ov), pappus (Pap), anther tube (An) and corolla (Co). The red-dotted section shows exemplarily the expected site of pollen (Po) recognition on the papilla (Pa) cell area of the forked stigma (f. St). (**b**) Lactic acid fixation of self-pollinated (SP) TkMS2. The germinated pollen tube shows strong aniline blue staining on the wall at the apex (white arrows). (**c**) SP of TkMS2. The germinating pollen tube can be traced to the papilla cells, where the pollen tube appears swollen and no pollen tube is observed within the stigma (double-headed arrows). (**d**,**e**) Lactic acid fixation of cross-pollinated (CP) TkMS2xTkMS3. (**d**) Several pollen grains are visible on the papilla surface, with two pollen tubes emerging and penetrating the stigma (white arrow). (**e**) Pollen tube growth between papilla cells. (**f**) The germinated pollen tube of the compatible pollination can be traced from the pollen and penetrates a short distance into the stigma before the signal disappears (white arrow). Scale bars in (**a**, bottom left) = 250 µm, (**a**, red-dotted section) = 50 µm and in (**b**–**f**) = 10 µm. Staining time (t) in (**b**) = 60 min, (**c**) = 24 h, (**d**) = 30 min, (**e**) = 40 min, (**f**) = 24 h. Abbreviations (**b**–**f**): Pa, papilla cells; Po, pollen; St, stigma.

**Figure 3 plants-10-00555-f003:**
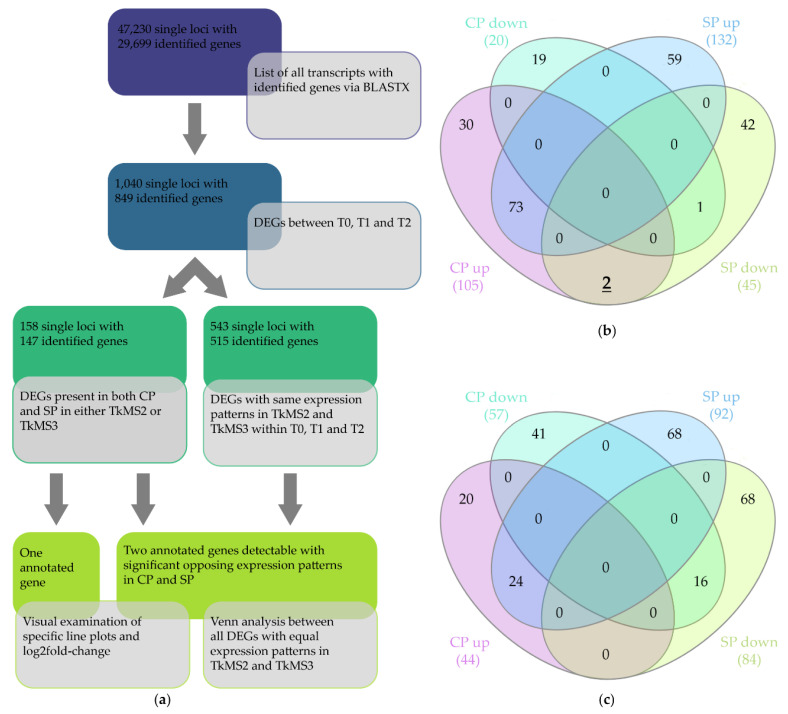
Summary of the filtering steps used to identify differentially expressed genes (DEGs) in *T. koksaghyz* pistil tissue during cross-pollination (CP) and self-pollination (SP). All pooled time points (T0, T1 and T2) were used for both varieties (TkMS2 and TkMS3). DEGs were defined according to the following criteria: FDR ≤ 0.05 and log_2_ fold-change ≥1 or ≤−1. (**a**) Workflow of the filtering strategy, with DEGs displaying multiple significant differences in expression pattern counted once. One annotated DEG was derived from the visual analysis of genes depicting significant log_2_ fold-changes during CP and SP in either variety (TkMS2 or TkMS3), and two DEGs were identified with the same expression pattern in TkMS2 and TkMS3 at all three time points (T0, T1 and T2) but differing between CP and SP. (**b**,**c**) Venn diagrams of DEGs depicting the expression profile during CP or SP in the *T. koksaghyz* varieties TkMS2 and TkMS3. The time points T1 vs. T0 (**b**) and T2 vs. T1 (**c**) are compared. Genes showing an opposing expression profiles are underlined and bold.

**Figure 4 plants-10-00555-f004:**
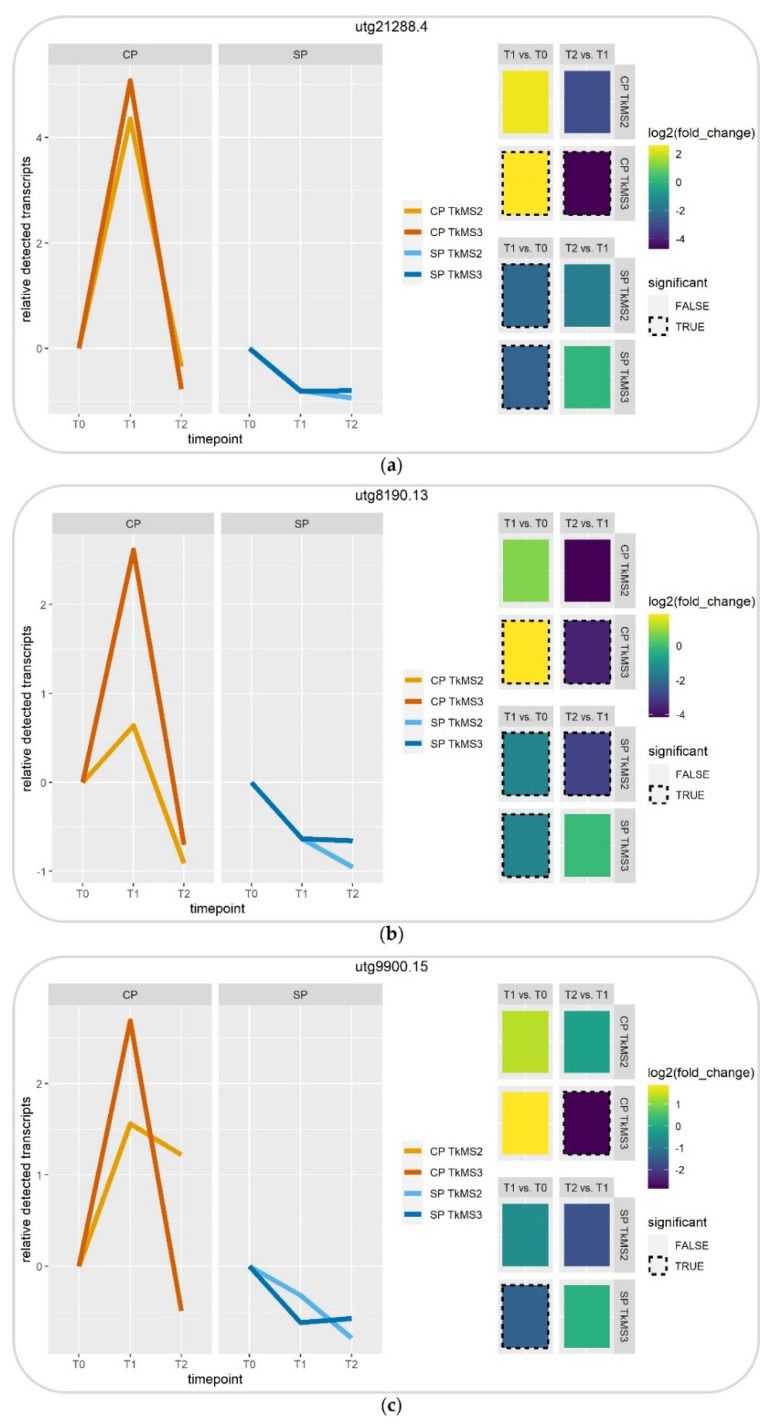
Visualization of relative transcript levels and log_2_ fold-changes of selected differentially expressed genes (DEGs) at different time points of cross-pollination (CP) and self-pollination (SP). The x-axes in the left panel show the pooled time points (T0, T1 and T2) and the y-axes show the transcript levels relative to T0 during the pollen–pistil interaction. The cross-pollination TkMS2xTkMS3 (CP TkMS2) is shown in light orange and TkMS3xTkMS2 (CP TkMS3) in dark orange. The self-pollination TkMS2 (SP TkMS2) is shown in light blue and TkMS3 (SP TkMS3) in dark blue. The right panel shows the corresponding log_2_ fold-changes, with significant values marked with a black dotted rectangle. The three selected DEGs are utg21288.4 (**a**), utg8190.13 (**b**) and utg9900.15 (**c**).

**Figure 5 plants-10-00555-f005:**
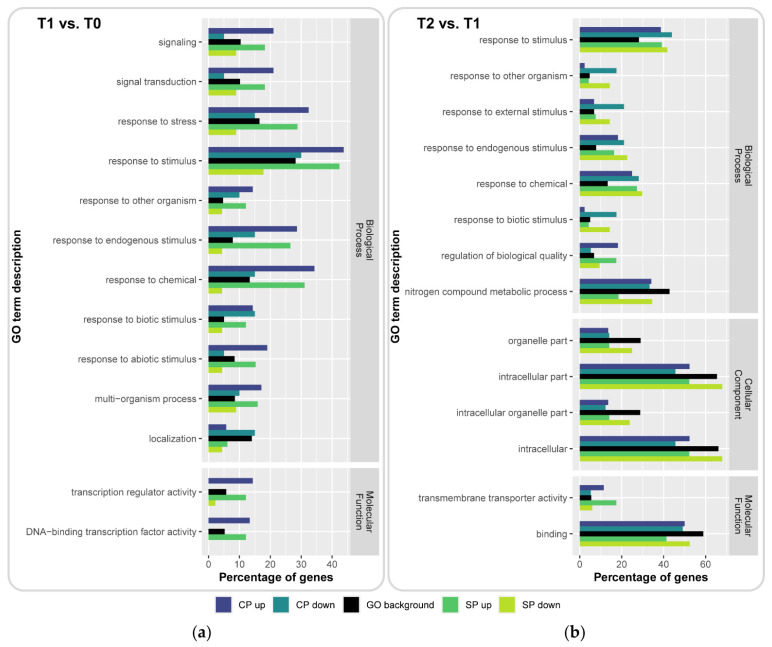
Enriched Gene Ontology (GO) terms (y-axis) based on significant *p*-values (≤ 0.001) of differentially expressed genes (DEGs) summed for TkMS2 and TkMS3 in the cross-pollination (CP) and self-pollination (SP) sets used for Venn diagram analysis, focusing on the comparisons (**a**) T1 vs. T0 and (**b**) T2 vs. T1 against the time point-independent GO background (with all GO terms associated with identified genes). Relative abundance (x-axis) of DEGs are grouped based on their expression pattern (CP upregulated in dark blue, CP downregulated in light blue, SP upregulated in dark green, SP downregulated in light green) compared with the GO background. Enriched GO terms are arranged by category of biological process, cellular component and molecular function, and were analyzed using WEGO.

**Table 1 plants-10-00555-t001:** Selected DEGs and their BLASTX sequence identity values.

Selected DEGs	Source	BLASTX e-Value ^1^	Percent Identity	Description	Abbreviation
utg21288.4	Venn/Visual analysis	0.0	82.67%	leucine-rich repeat extensin-like protein 4 [*Lactuca sativa*]	*LRX4*
utg8190.13	Venn/Visual analysis	9 × 10^−150^	99.55%	tubulin beta chain-like [*Lactuca sativa*]	*TUBB*
utg9900.15	Visual analysis	0.0	91.77%	probable xyloglucan endotransglucosylase/hydrolase protein 33 [*Lactuca sativa*]	*XTH33*

^1^ Based on non-redundant protein sequences (nr) database.

## Data Availability

The data presented in this study are available in Appendix A.

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
