# Peer review of "Microscopic and Transcriptomic Analysis of Pollination Processes in Self-Incompatible Taraxacum koksaghyz"

_plants, 2021, doi:10.3390/plants10030555_

Round 1

Reviewer 1 Report

This is a well written and structured manuscripr and I suggest to accept it in the present form.

Author Response

Thank you very much for your kind words and for accepting the manuscript in the present form.  

Reviewer 2 Report

The manuscript is well-written, interesting, and informative.  I have a few minor questions that the authors should be able to address, easily.

I had some difficulty understanding Figure 2, in part because I was initially getting pappus (PAP) confused with references to "papilla cells". There are some terms and references in the caption of Figure 2 that were not well defined. It would be helpful to have another schematic in Figure 2 that helps explain all anatomical parts referenced in Figure 2. What is the line in the lower-left part of Figure 2 (a), is that a scale bar? 

The authors discuss "undesirable" attributes of SI in terms of crop breeding and crop domestication, but there are many important SI crops and there are many breeding methods, procedures, and theories that have been developed for SI plants (and animals). I agree that it would be useful if we can control SI for the breeding and development of crops, but I don't think that the authors need to use "undesirable attributes of SI" as a sole justification for the work. 

I would like authors to discuss DUF4428 protein DEG. It seems that the authors dismiss  DUF4428 because they want to focus on DEGs upregulated during CP as explained lines 392-402? This does not seem like a good reason to dismiss a potentially important observation. Genes encoding DUF proteins have been directly implicated in the control of GSI in grasses (Lolium), so I thought this warrants some discussion.

The authors indicate that the "strongest evidence for an SSI mechanism is the absence of pollen on the papilla cells and the tip swelling" (Lines 387-388). Is there as citation for this?  A little bit more explanation for the evidence of SSI (lines 386-389) would be helpful. 
